# Fibroblast growth factors (FGFs) prime the limb specific *Shh* enhancer for chromatin changes that balance histone acetylation mediated by E26 transformation-specific (ETS) factors

**Silvia Peluso[1], Adam Douglas[1], Alison Hill[1], Carlo De Angelis[1], Benjamin L Moore[1], Graeme Grimes[1], Giulia Petrovich[1], Abdelkader Essafi[2], Robert E Hill[1]\***

[1]MRC Human Genetics Unit, Institute of Genetics and Molecular Medicine, University of Edinburgh, Edinburgh, United Kingdom; [2]School of Cellular and Molecular Medicine, Faculty of Biomedical Sciences, University of Bristol, Bristol, United Kingdom

**Abstract** Sonic hedgehog (*Shh*) expression in the limb bud organizing centre called the zone of polarizing activity is regulated by the ZRS enhancer. Here, we examine in mouse and in a mouse limb-derived cell line the dynamic events that activate and restrict the spatial activity of the ZRS. Fibroblast growth factor (FGF) signalling in the distal limb primes the ZRS at early embryonic stages maintaining a poised, but inactive state broadly across the distal limb mesenchyme. The E26 transformation-specific transcription factor, ETV4, which is induced by FGF signalling and acts as a repressor of ZRS activity, interacts with the histone deacetylase HDAC2 and ensures that the poised ZRS remains transcriptionally inactive. Conversely, GABPα, an activator of the ZRS, recruits p300, which is associated with histone acetylation (H3K27ac) indicative of an active enhancer. Hence, the primed but inactive state of the ZRS is induced by FGF signalling and in combination with balanced histone modification events establishes the restricted, active enhancer responsible for patterning the limb bud during development.
DOI: https://doi.org/10.7554/eLife.28590.001

**\*For correspondence:**
Bob.hill@igmm.ed.ac.uk

**Competing interests:** The authors declare that no competing interests exist.

## Introduction

Spatial specific gene expression is fundamental to controlling cell identity in embryonic tissue. Early in the mesenchyme of the developing mammalian limb bud there are no observable morphological differences or histological boundaries; nevertheless, the limb bud is initially polarized along the anterior-posterior axis (*Tickle, 2015*) establishing a specialized compartment of cells at the posterior margin called the zone of polarizing activity (ZPA). The function of the ZPA is the expression of *Shh*, which operates as a morphogen and a mitogen to coordinate digit formation by integrating growth with digit specification during limb development (*Towers et al., 2008*; *Zhu et al., 2008*). A highly conserved ~780 bp enhancer called the ZRS controls the spatiotemporal expression of the *Shh* gene in the ZPA of both the fore and hind limbs (*Lettice et al., 2002*; *2003*; *Sagai et al., 2005*). The ZRS lies in an intron of the ubiquitously expressed *Lmbr1* gene at a distance of 1 Mb from the *Shh* gene in human. Well over 20 different point mutations occurring in the ZRS are associated with the misregulation of *Shh* and consequently, to limb skeletal defects. These include preaxial polydactyly type 2 (PPD2), triphalangeal thumb polysyndactyly, syndactyly type IV, and Werner's mesomelic syndrome, collectively referred to as ZRS-associated syndromes (*Lettice et al., 2003*, *Lettice et al., 2008*;

**eLife digest** As an animal embryo develops, specific genes need to be switched on and off at the right time and place to ensure that the embryo's tissues and organs form properly. Proteins called transcription factors control the activity of individual genes by binding to regions of DNA known as enhancers. Changes in the way DNA is packaged inside cells can affect the ability of transcription factors to access the enhancers, and therefore also influence when particular genes are switched on or off.

Sonic hedgehog (or *Shh* for short) is a gene that helps to control various aspects of development including the formation of the limbs and brain. The limb forms from a structure in the embryo referred to as the limb bud. An enhancer called ZRS regulates the precise position within the limb bud where the *Shh* gene is active in a region designated as the "zone of polarizing activity". Yet, it was not known how the enhancer is controlled to ensure this pattern is achieved. Peluso et al. investigated the events that lead to ZRS becoming active in mice embryos.

The experiments show that the ZRS enhancer exists in three different states in cells across the limb bud: poised, active and inactive. The enhancer is poised in a broad region of the limb bud in cells that are potentially able to switch on the *Shh* gene. Proteins called fibroblast growth factors drive the enhancer to enter this poised state by altering the way the DNA containing the enhancer is packaged in the cell. Specific transcription factors are able to bind to the poised enhancer and it is the balance between these different transcription factors that activates the enhancer in the zone of polarizing activity. Furthermore in the region of the limb bud where the fibroblast growth factors are not present the ZRS is inactive.

These findings show that fibroblast growth factors, in combination with other changes to the ZRS enhancer, restrict the area in which the enhancer is active to a particular region of the limb bud. Differences in enhancer elements are known to underlie a range of inherited characteristics and may influence whether an individual develops many common diseases. In the future, investigating how cells control the activity of enhancers may provide clues to identifying new targets for drugs to treat some of these diseases.

DOI: https://doi.org/10.7554/eLife.28590.002

*Farooq et al., 2010*; *Furniss et al., 2008*; *Gurnett et al., 2007*; *Semerci et al., 2009*; *Wieczorek et al., 2010*).

The regulatory mechanisms that direct gene expression to embryonic compartments are not well established in mammalian development; therefore, to understand the events that occur, it is crucial to investigate the activation of spatiotemporal specific enhancers during this process. We previously showed that members of the E26 transformation-specific (ETS) transcription factor family (*Sharrocks, 2001*) are involved in the spatial pattern of *Shh* expression (*Lettice et al., 2012*). Occupancy at multiple ETS sites, which bind the factors GABPα and ETS1, regulates the position of the *Shh* expression boundary in the limb, thus defining the ZPA. In contrast, binding sites for ETV4 and ETV5 in the ZRS, when occupied, repress ectopic *Shh* expression outside the ZPA. A single base pair change is able to subvert this normal developmental process to give rise to skeletal abnormalities. For example, two different human PPD2 point mutations generate additional ETS binding sites, thereby, de-repressing expression of *Shh* in the anterior limb bud. The balance between binding of the activators and repressors is crucial for normal *Shh* expression and the mutations that disrupt this balance result in skeletal defects.

Analysis of developmental gene regulation must also take into account the mechanisms of long-range enhancer/promoter interactions. Previously, we showed that the ZRS contains two domains with distinct activities; one domain, the 5' end of the ZRS (472 bp), directs the spatiotemporal activity and the second domain, the 3' half (308 bp), is required to mediate activity over long genomic distances (*Lettice et al., 2014*). Additionally higher-order chromatin conformational changes that occur in the *Shh* locus play a role in gene expression. Elevated frequencies of *Shh*/ZRS co-localization were observed only in the *Shh* expressing regions of the limb bud (*Amano et al., 2009*), in a conformation consistent with enhancer-promoter loop formation (*Williamson et al., 2016*). However, the

domain between *Shh* and ZRS is highly compacted in all tissues and developmental stages analysed independent of *Shh* expression.

Here, we investigate the stepwise events that mediate the spatiotemporal activation of an enhancer during development. We demonstrate that even though the activity of the ZRS is restricted to the ZPA, it retains features of a poised enhancer along the full distal portion of the limb bud composed of the mesenchymal cells of the progress zone; whereas, H3K27ac is enriched just in the distal-posterior limb region. We show that fibroblast growth factor (FGF) signalling plays a key role in priming the ZRS as indicated by an increase in H3K4me1 at the enhancer. In addition, we show the mechanism of acetylation/deacetylation that GABPα, an activator, and ETV4, a repressor, employ to restrict and promote ZRS activation in the distal limb bud.

## Results

### Modifications of chromatin at the ZRS differ in different regions of the limb bud

In order to characterize specific features of the *Shh* limb enhancer, the ZRS, in different regions of the developing limb, we microdissected the limb bud at embryonic day 11.5 (E11.5) into several defined segments. Firstly, we examined the distal region which contained the specialized epithelial structure called the AER (apical ectodermal ridge), the mesenchyme of both the progress zone and the ZPA and, in addition, the proximal region which contained the shank of the limb bud (*Figure 1A*). Chromatin immunoprecipitation analysed by quantitative PCR (ChIP-qPCR) was performed on the dissected limb tissue for the two modified histones, H3K4me1 and H3K27ac (*Figure 1B*). Previous studies demonstrated that these modifications are markers for enhancer activity. H3K4me1 is a predictive chromatin signature for both poised and active enhancers in the human genome (*Heintzman et al., 2007*), but in association with H3K27ac, these mark active regulatory elements (*Rada-Iglesias et al., 2012*; *Cotney et al., 2012*). We found specific enrichment of H3K4me1 at the ZRS in the distal region of the dissected E11.5 limb buds, which was appreciably lower in the proximal region of the limb buds. Further dissections of the distal limb bud into anterior and posterior halves enabled us to discern more precisely the location of the H3K4me1 at the ZRS (*Figure 1B*). Even though *Shh* was not expressed in the anterior region of the limb bud, the H3K4me1 mark was enriched in both dissected halves suggesting that the ZRS was poised across the distal compartment of the limb.

H3K27ac, in contrast, was differentially enriched in the distal mesenchyme which was greater in the posterior portion of the limb bud. To investigate acetylation of histone H3 across the large 770 bp enhancer, two sets of PCR primers were used to amplify each end of the ZRS. The ZRS encompasses two distinct regulatory activities; one residing in the 5′ half driving spatiotemporal expression and one residing in the 3′ half mediating long-range activation of the *Shh* gene (*Lettice et al., 2014*) (*Figure 1C*). Analysis of each half of the enhancer after ChIP identified differential acetylation of H3 across the ZRS. PCR primers complementary to the 5′ spatiotemporal half (5′ST primers) of the ZRS showed an increase in H3 acetylation compared to the 3′ long-range (3′LR primers) end. H3K27ac was associated with the 5′ end of the ZRS and was enriched in the distal region of the limb bud. H3K27ac was, therefore, associated with the distal, posterior quadrant of the limb bud consistent with the presence of the ZPA where the *Shh* gene is transcriptionally active (*Figure 1A*). Hence, in the case of the ZRS, the enhancer is poised throughout the posterior distal mesenchyme; further events occur to activate the enhancer for productive transcription in the ZPA.

We next assessed the chromatin state of the ZRS in the distal region of the limb bud using ChIP coupled with microarray called ChIP-on-chip. The ChIP analysis using antibodies to H3K4me1 and H3K27ac highlighted the specificity of the histone modifications over the ZRS (*Figure 1D*). The pattern of the H3K4me1 covered the extent of the ZRS; whereas, the pattern of H3K27ac, as predicted by the PCR primers, was not centred over the ZRS but was skewed toward the 5′ side. Thus, the ZRS exists in differential chromatin states in the limb depending on position and gene activity.

### A limb-derived cell line shows ZRS activation and *Shh* induction

To investigate the dynamics of *Shh* regulation in relation to the ZRS, we generated immortalized cell lines from early limb bud mesenchyme (*Williamson et al., 2012*). RNA-seq analysis of the 14Fp cell

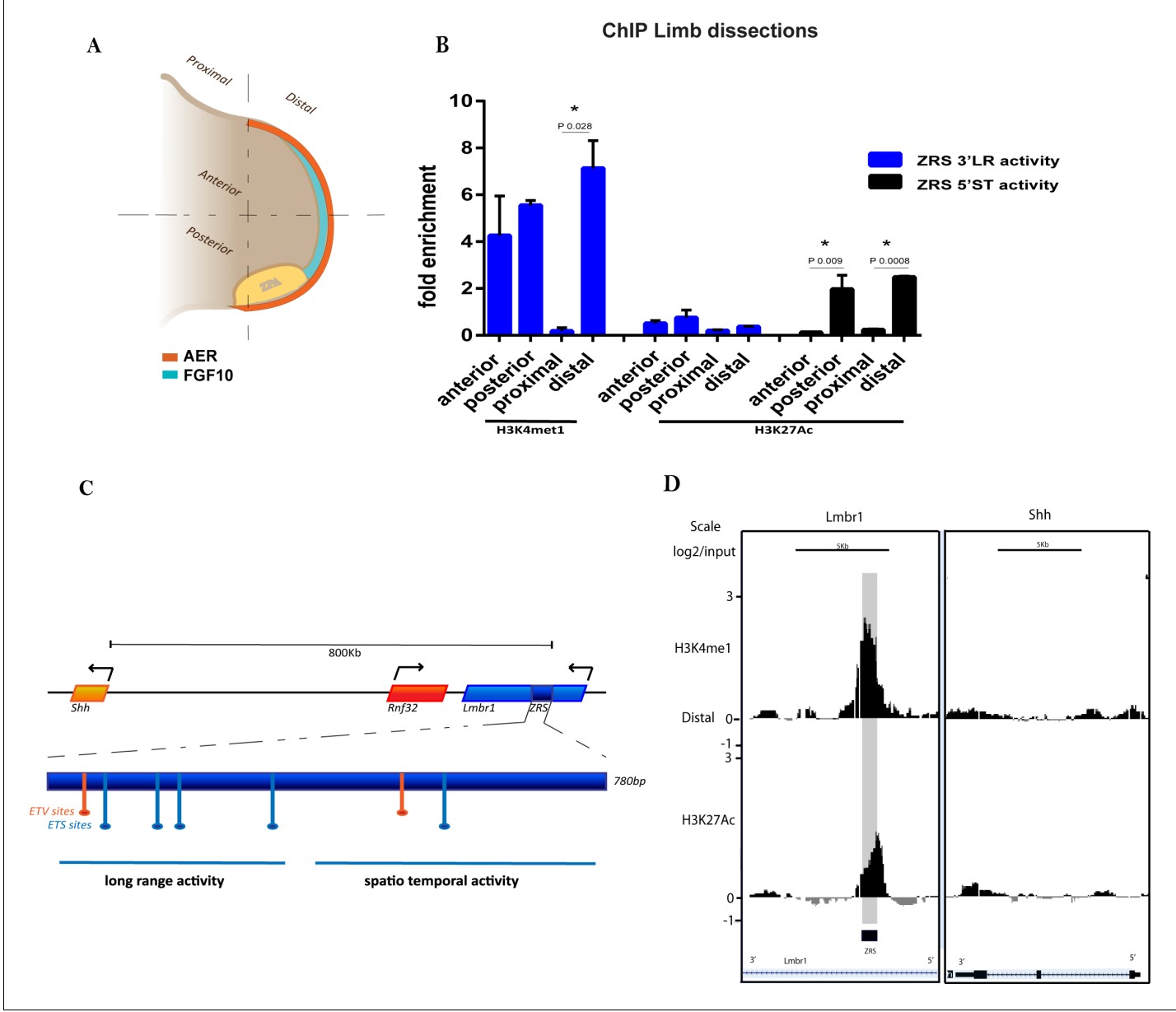

**Figure 1.** H3K4me1 and H3K27ac distribution over the ZRS in embryonic day 11.5 (E11.5) limb buds. (**A**) Representation of an E11.5 mouse limb bud. The apical ectodermal ridge (orange), the ZPA (yellow) and FGF10 (cyan) are responsible for the anterior/posterior limb patterning and for directing the proliferation of the distal portion of the limb. The limb buds were dissected into anterior-posterior or distal-proximal regions. (**B**) Chromatin from E11.5 dissected limb buds tissue was enriched by chromatin immunoprecipitation (ChIP) for H3K4me1 and H3K27ac histone modifications. DNA was quantified by quantitative PCR. Means of fold enrichment over nonspecific IgG recoveries and ±SEM from two independent experiments are plotted. (**C**) Schematic of the ZRS (dark blue box) lying in intron 5 of *Lmbr1* (blue box), 800 kb away from *Shh* gene. Within the ZRS (dark blue box) five ETS binding sites (light blue) and two ETV binding sites (orange) are highlighted. The primers used to evaluate the ChIP experiments are localized over the long-range activity and the spatiotemporal activity sequences, both indicated with a blue line (complete list of oligos in ***Supplementary file 1***). (**D**) ChIP-on-chip analysis of distal mesenchyme from two biological replicates of E11.5 limb buds using antibodies to two different histone modifications (H3K4me1 and H3K27ac). Data for two different genomic regions, the fifth intron of *Lmbr1* gene and the *Shh* gene, are shown. The y axis is the $\log_2$ for each ChIP/input DNA and the x axis represents a segment of DNA. The DNA region containing the ZRS is highlighted by the grey shading. As controls, the whole of the Shh coding region plus promoter (Shh) is shown.

DOI: https://doi.org/10.7554/eLife.28590.003

line (cells derived from the distal/posterior part of the limb at E 11.5) shows that many of the key genes found in the posterior limb bud are expressed with a notable exception being *Shh* (*Figure 2—figure supplement 1A*). We performed ChIP-on-chip using antibodies against H3K4me1 and H3K27ac which revealed that, H3K4me1 marked the full extent of the ZRS in the cell line; whereas, H3K27ac was not enriched, demonstrating that the ZRS resides in a poised state reflecting the origin of the cell line from the distal mesenchyme of the limb bud (*Figure 2A* and *Figure 2—figure supplement 1A*).

The poised state of the ZRS in the cells indicated that *Shh* inactivation was due to the lack of specific factors that are responsible for fully activating the ZRS. Previous attempts to activate *Shh* expression in limb bud-derived cells using cocktails of known developmental activators such as FGFs and retinoic acid or transfection with HoxD genes have shown that *Shh* is refractory to activation (*Kimura and Ide, 1998*). Our attempts with known *Shh* activators confirmed these observations (*Figure 2—figure supplement 2A*); however, the drug trichostatin A (TSA), a histone deacetylase (HDAC) inhibitor, stimulated expression (*Figure 2—figure supplement 2B–C*). *Shh* expression was detectable within 6 hr of treatment (*Figure 2B*) reaching a maximum at ~24 hr. To further investigate *Shh* activation in the cell line, we carried out a TSA time course to assay H3K27ac enrichment at the ZRS. H3K27ac enrichment was observed over both the ZRS and the *Shh* promoter at 16 hr of treatment, reaching a maximum at 24 hr (*Figure 2C*), accordingly, with the *Shh* expression time course (*Figure 2B*). The fold enrichment of H3K27ac in the limb buds (*Figure 1B*) and cell lines (*Figure 2C*) seemed dramatically different; therefore, enrichment of H3K27ac was directly compared between the limbs buds and the cells in the same experiment showing that the magnitude of enrichment is comparable (*Figure 2—figure supplement 3A*). In the cell lines, the correlation between *Shh* induction of expression and recruitment of H3K27ac suggests that the ZRS is involved in the activation of the transcriptional process. In addition H3K27ac was more highly associated with the region involved in the spatiotemporal activity of the ZRS (as shown using the 5'ST primers). Two control mouse cell lines were used to investigate the specificity of *Shh* activation in the 14Fp cells. Firstly, the mouse ES cell line, E14, which in response to retinoic acid induces *Shh* expression, and secondly, an immortalized mesenchymal cell line from the embryonic mandible at E11.5, called the MD cell line, were used. In both cell lines, the histone mark, H3K4me1, was not enriched over the ZRS, and TSA treatment did not lead to *Shh* induction after 24 hr of treatment (*Figure 2—figure supplement 3B–C*).

Extragenic transcription sites correlate with active regulatory elements and in accord are occupied by RNA Pol II (*De Santa et al., 2010*; *Kim et al., 2010*). Genome-wide studies highlighted Pol II occupancy and the synthesis of noncoding transcripts at active enhancer sites, the roles of which, so far, remain unclear. In order to confirm the limb specific enhancer activation, Pol II ChIP-qPCR was carried out in limb cells after TSA treatment. In the untreated 14Fp cell line, even though the enhancer is in a poised state, Pol II was not appreciably enriched at the *Shh* promoter nor at the ZRS (*Figure 2D*). After activation with TSA, Pol II was detected at both sites confirming the association of Pol II recruitment to the enhancer and the promoter, further suggesting that TSA mediated *Shh* activation in 14Fp involves the ZRS. Thus, this cell line derived from early limb bud mesenchyme showed that the ZRS could be induced to undergo modifications consistent with enhancer activation and concomitantly, *Shh* expression was activated. To further establish the role for ZRS in the activation of Shh expression in the cell line, the interaction of the ZRS with its target promoter was analysed.

## Promoter-enhancer contacts are established during *Shh* gene activation

*Shh* gene activation is linked to an increase in the level of the H3K27ac histone mark over the ZRS. In order to investigate the reorganization of chromatin structure after TSA treatment and to confirm the involvement of the ZRS in *Shh* activation induced by TSA, circularized chromosome conformation capture (4C-seq) (*Stadhouders et al., 2013*) analysis was carried out at 18 and 24 hr after TSA treatment (*Figure 2—figure supplement 3D–F*). Evidence of increased co-localization of the ZRS and the *Shh* gene in the expressing region of the limb bud was demonstrated previously by FISH and 3C analysis (*Amano et al., 2009*, *Williamson et al., 2016*). 4C-seq analysis in the 14Fp cell line (*Figure 3A*) showed marked and highly significant ZRS–*Shh* interactions in the TSA treated cells. The interaction between the *Shh* gene and the ZRS was confirmed by 3C-qPCR (*Figure 3—figure supplement 1E*).

The fragment containing the promoter site of *Shh* co-localizes with the ZRS and was detected at 18 and at 24 hr after TSA treatment (TSA minus: q < $1.5 \times 10^{-5}$; TSA$^{18hr}$: q-value <$5 \times 10^{-10}$; TSA$^{+24hr}$:

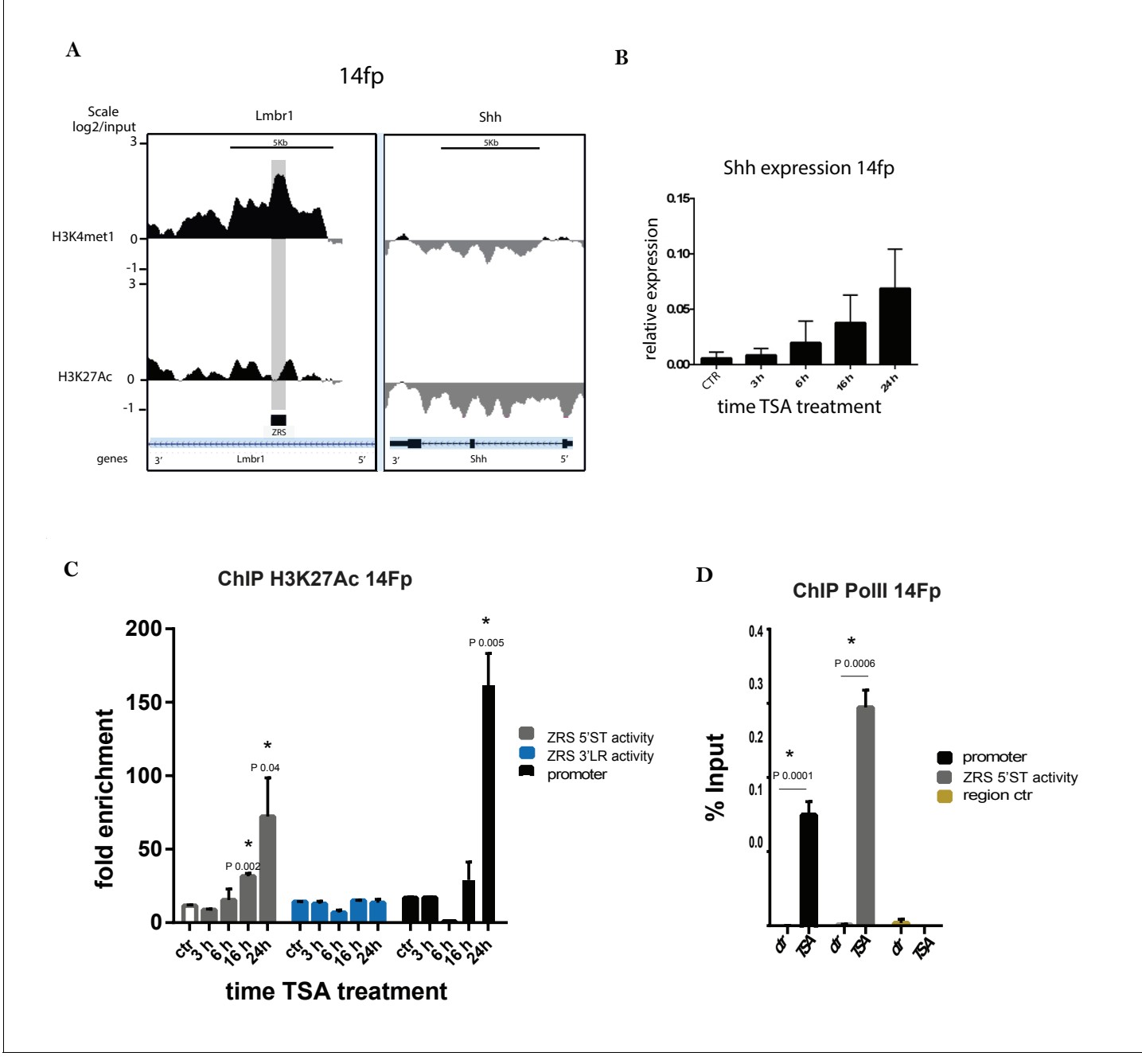

**Figure 2.** Trichostatin A (TSA) treatment activates *Shh* in a limb-derived cell line (14Fp). (**A**) Chromatin immunoprecipitation (ChIP) from two biological replicates of E11.5 limb-derived cell line (14Fp cell line) using antibodies to two different histone modifications (H3K4me1 and H3K27ac) analysed by hybridizing to tiling microarrays. Summary is presented using two different genomic regions, the y axis is log$_2$ for each ChIP/input DNA and the x axis represents a segment of DNA from the microarray. The DNA region containing the ZRS is highlighted by the grey shading. As controls, the whole of the *Shh* coding region plus promoter is shown. (**B**) Time course of the expression of *Shh* in E11.5 limb-derived cell line after TSA treatment detected by quantitative reverse transcriptase PCR. The Shh levels were evaluated relative to control and normalized to glyceraldehyde 3-phosphate dehydrogenase expression levels. Data points represent the average of triplicate determinations ± SEM. (**C**) Chromatin from 14Fp was harvested 3, 6, 18, and 24 hr after TSA treatment or 24 hr with DMSO as control (ctr). Shown are results from ChIP analysis using anti-H3K27ac antibody. Enrichment of H3K27ac at the 5' spatiotemporal (5'ST) (grey), 3' long range (3'LR) (blue) and promoter (black) was detected by quantitative PCR and represented as mean of fold enrichment/background (IgG) ± SEM over three biological replicates; a negative control region was analysed and did not give an appreciable signal (data not shown). (**D**) Shown are results from ChIP analysis using anti-RNA Pol II antibody after 24 hr of TSA treatment. Indicated are the *shh* promoter (black), the 5'ST (grey) and control region (yellow). Recovered DNA sequences were quantified as percentage of input and ±SEM from two independent experiments and are plotted. The IgG did give no detectable signal.

*Figure 2 continued on next page*

*Figure 2 continued*

DOI: https://doi.org/10.7554/eLife.28590.004

The following figure supplements are available for figure 2:

**Figure supplement 1.** RNA-seq analysis of the 14Fp cell line and of the distal limb bud.
DOI: https://doi.org/10.7554/eLife.28590.005
**Figure supplement 2.** Induction of *Shh* expression in the 14Fp cell line.
DOI: https://doi.org/10.7554/eLife.28590.006
**Figure supplement 3.** Specificity of *Shh* expression and efficiency of 4C library preparation after TSA induction in the 14Fp cell line.
DOI: https://doi.org/10.7554/eLife.28590.007

$q < 7.8 \times 10^{-35}$); whereas, significant contact in the fragment containing the promoter was undetectable in the untreated cells. *Shh*/ZRS proximity in the nucleus occurs regardless of whether the gene or enhancer is active (*Williamson et al., 2016*); however, in TSA treated cells reorganization in chromatin structure occurs due to activation of the ZRS increasing interactions with the promoter.

## FGF signalling is responsible for priming the ZRS

At an early stage of limb development, *Fgf10* expression in the distal limb bud mesenchyme is important for both limb bud outgrowth and induction of FGF signalling from the AER (*Ohuchi et al., 1997*) which, in turn, maintains the cells of the progress zone. Subsequently, the FGFs function to maintain *Shh* expression in the ZPA (*Laufer et al., 1994*; *Niswander et al., 1994*; *Crossley et al., 1996*; *Vogel et al., 1996*; *Ohuchi et al., 1997*). We tested the hypothesis that localized FGF expression is correlated with the poised state of the ZRS. Firstly, the 14Fp cell line pre-treated with nintedanib (NIN), a potent broad spectrum inhibitor of FGFR1/2/3, VEGFR1/2/3, and PDGFRα/β (*Hilberg et al., 2008*), showed a significant reduction in *Shh* after TSA treatment (*Figure 4A*). Previous studies show that an increase in AER-FGF levels leads to gradual repression of *Grem1* in the distal mesenchyme as part of an inhibitory feedback loop (Fgf/Grem1 loop) (*Verheyden and Sun, 2008*) and promotes expression of *Etv4* (*Mao et al., 2009*). Hence, as control for the efficiency of the FGF inhibition, *Grem1* and *Etv4* levels were evaluated (*Figure 4B*) after 4 hr incubation with NIN or after 4 hr NIN plus FGF8/10 incubation. Since NIN can also inhibit VEGF and PDGF receptors, another FGFR inhibitor, BGJ398 (BGJ), was also tested for *Grem1* and *Etv4* expression. Increased levels of *Grem1* were observed after 4 hr of NIN and BGJ incubations while *Etv4* levels were reduced. The FGF8/10 treatment was sufficient to restore the original levels of both *Grem1* and *Etv4*. To investigate the action of FGF signalling at the ZRS, ChIP for H3K4me1 in the limb specific cell line exposed to NIN was performed (*Figure 4C*). As control, a region of the first intron of *Rbm33*, a neighbouring gene, which displays open chromatin coincident with a peak of H3K4me1 was examined. The H3K4me1 enrichment over the ZRS was dramatically reduced after inhibition of FGF activity and TSA treatment did not rescue the presence of this histone modification; whereas, *Rbm33* intron 1 was not significantly affected. The same effect was caused by BGJ on H3K4me1 enrichment (*Figure 4—figure supplement 1D–E*), and no differences were observed in comparison with NIN treatment. To assess whether the NIN or BGJ treatment would affect cell survival or cause other abnormality trypan blue staining was performed and no alterations were observed after 4 hr treatment (*Figure 4—figure supplement 1A*).

To determine if FGF plays a similar role in effecting the poised state of the ZRS in the embryo, we developed a short-term organ culture approach (*Havis et al., 2014*). Distal tips of E11.5 limb buds were dissected, maintained in media, and exposed to NIN for 4 hr to examine the state of endogenous ZRS. In agreement with the cell line results, the distal tips lost ZRS enrichment of H3K4me1 and higher levels of *Grem1* expression were observed (*Figure 4F–G*). On the other hand, when proximal dissections of E11.5 limb bud, where the enhancer is in an inactive state, are exposed to a combination of FGF8 and FGF10 the ZRS displays the poised state (*Figure 4H*, *Figure 4—figure supplement 1F*) showing enrichment of H3K4me1. These data suggest that FGF signalling has a key role in priming and maintaining the ZRS as a poised enhancer in the distal mesenchyme of the limb bud, delineating the boundaries where *Shh* can be potentially expressed. Trypan blue staining was performed on limb dissections by following a reported protocol with slight modifications

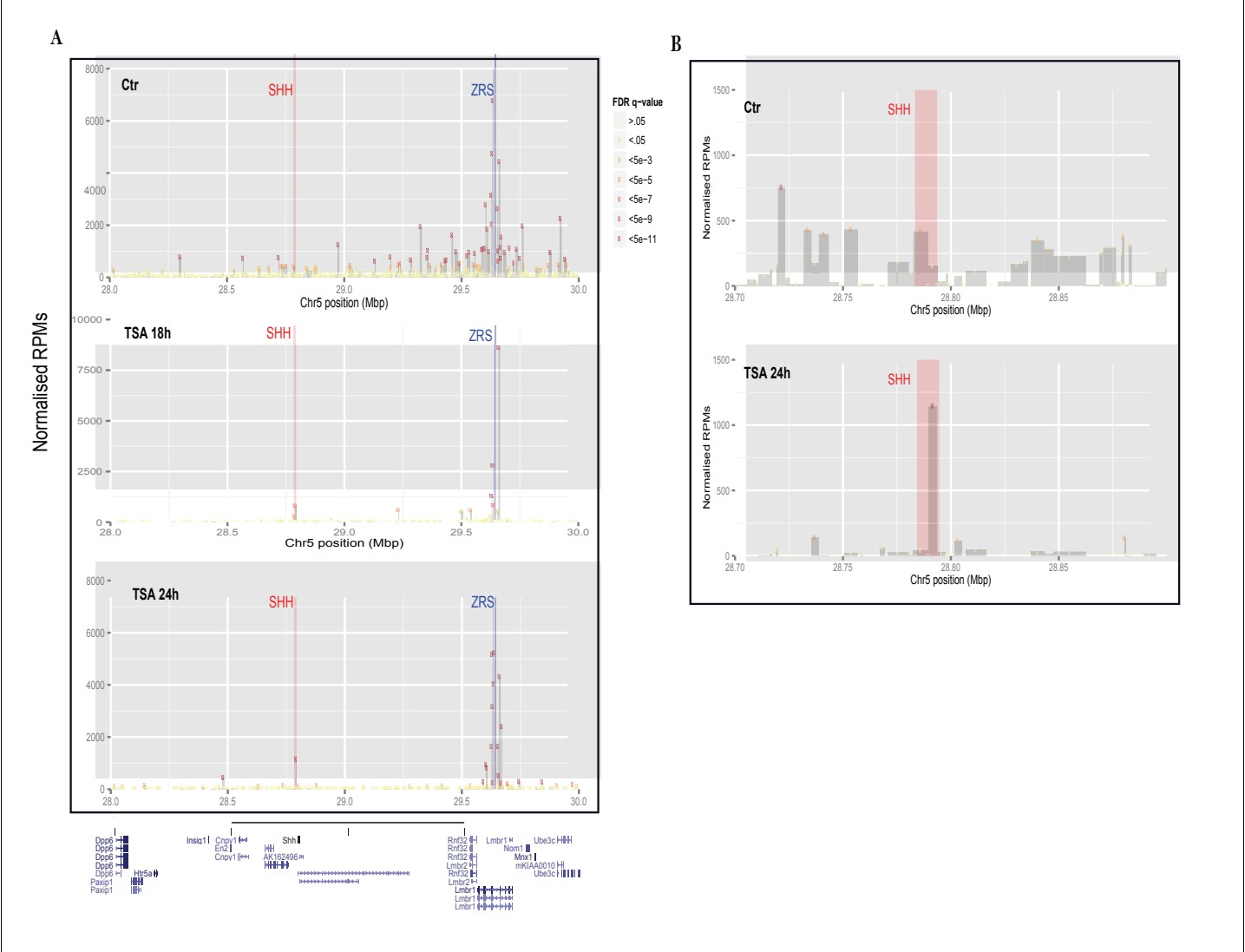

**Figure 3.** Trichostatin A (TSA) treatment induces a ZRS–SHH interaction. (**A**) The profile of 4C-seq at the ZRS locus in 14Fp cells. Control and TSA treatment after 18 and 24 hr are shown. ZRS shown as the enhancer (anchor) bait fragment along an approximately 2 Mb region of chromosome 5 (UCSC genome browser view of chr5:28,000,000–30,000,000 (mm9)). The x axis represents the position on chromosome 5 and the y axis the normalized reads as read per million sequences (RPMs). Only highly significant interactions are shown (false discovery rate [FDR] q-value $<5{\to}10^5$). The blue bar represents the location of the ZRS (bait) and the red bar represents the *Shh* gene. Each rectangle is a restriction fragment, the dots coloured at the top of each rectangle reflect the FDR q-value indicating the significance of the interaction (legend between A and B). (**B**) Focus on the Shh region (red bar in the zoomed-in view) shows the number of interactions of the bait region with SHH in both untreated and TSA treated (after 24 hr) samples.

DOI: https://doi.org/10.7554/eLife.28590.008

The following figure supplement is available for figure 3:

**Figure supplement 1.** Chromatin conformation capture (3C) analysis in 14Fp cells induced to expression *Shh*.

DOI: https://doi.org/10.7554/eLife.28590.009

(*Siddique, 2012*) (*Figure 4—figure supplement 1B*). No signs of increased cell death were observed after 4 hr of NIN treatment.

## GABPα activates *Shh* expression

The 14Fp cells retain the expression of some of the ETS genes (*Figure 2—figure supplement 1A*) which were shown to play a regulatory role at the ZRS (*Lettice et al., 2012*). One of these, GABPα, interacts with the ZRS in the limb bud and activates *Shh* expression. In addition others have

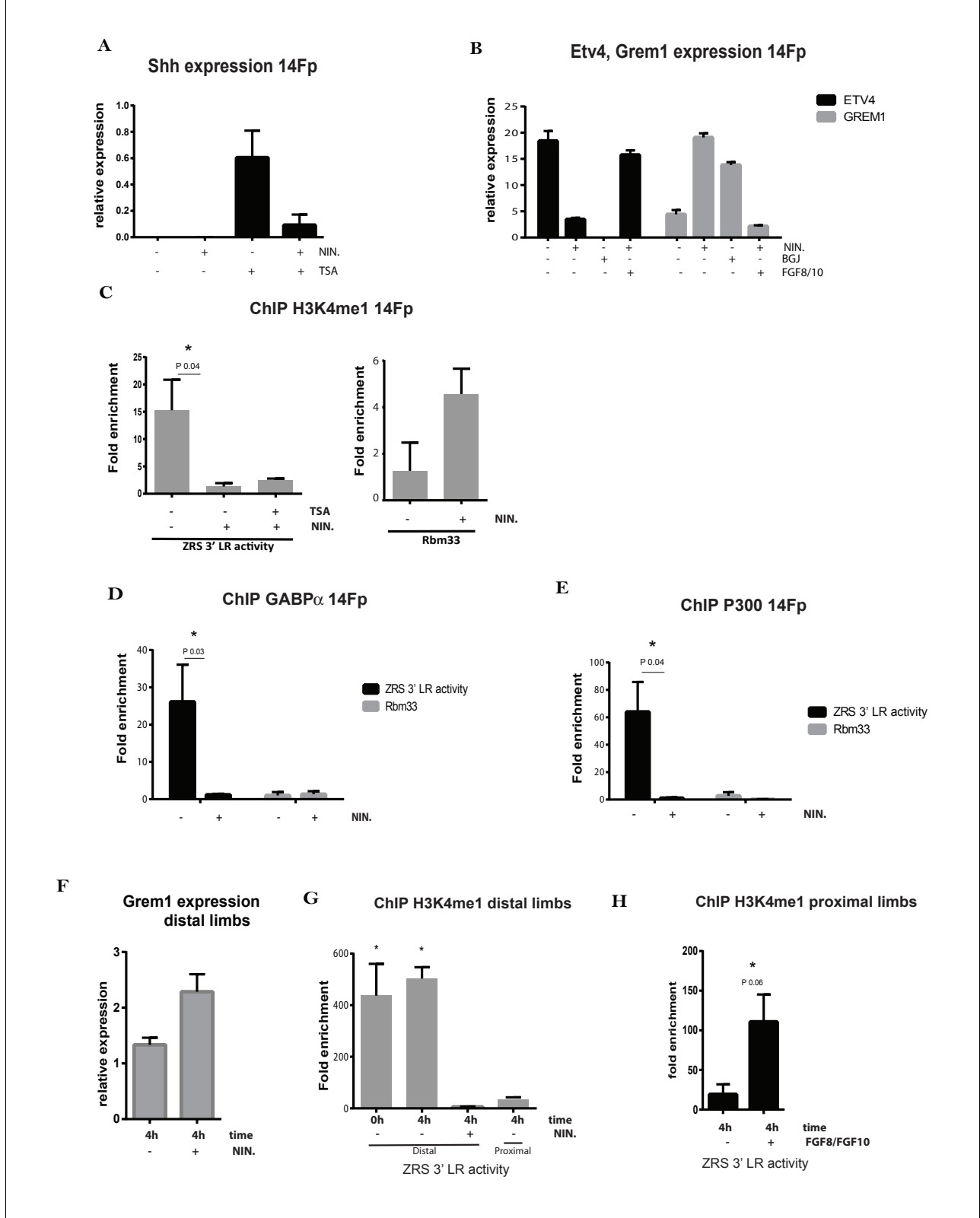

**Figure 4.** Fibroblast growth factors (FGFs) are crucial in priming *Shh* limb specific enhancer. (**A**) Quantitative reverse transcriptase (qRT)-PCR was used to detect the expression levels of *Shh* in 14Fp cell line after trichostatin A (TSA) and nintedanib (NIN) treatment. The *Shh* levels were evaluated relative to control and normalized to glyceraldehyde 3-phosphate dehydrogenase levels. Data points represent the mean of three biological replicates ± SEM. (**B**) qRT-PCR to detect the expression of *Grem1* (grey bars) and *Etv4* (black bars) after 4 hr of NIN, with or without supplement of FGF8/10 for 6 hr. Also *Figure 4 continued on next page*

*Figure 4 continued*

shown is BGJ398 (BGJ) treatment. Data points represent the average of duplicate determinations ± SEM. (C) Chromatin from the 14Fp cell line was harvested after TSA and NIN treatment and ChIP for H3K4me1 was carried out. DNA was quantified by q-PCR using the ZRS 3' long range (3'LR) and *Rbm33* oligos. Data are represented as mean ± SEM of the fold enrichment over nonspecific IgG recoveries from two independent experiments. (D–E) Chromatin immunoprecipitation (ChIP) analyses after NIN treatment where performed to further analyse the enrichment of the transcription factor GABPα and P300 over the ZRS (black) and on a specific genomic control region, Rbm33 intron (grey). DNA was quantified by q-PCR. Mean (±SEM) of the fold enrichment over nonspecific IgG recoveries from two independent experiments is plotted. (F) qRT-PCR to detect the expression of *Grem1* in the distal limb bud after 4 hr of NIN treatment. (G) ChIP of distal and proximal limb tissue from limb buds using an antibody against H3K4me1. Distal limb tissue was treated for 4 hr with or without NIN. Proximal limb tissue was used as negative control. DNA was quantified by q-PCR and fold enrichment over nonspecific IgG recoveries using the ZRS 3'LR oligos and ±SEM from two independent experiments were plotted. (H) H3K4me1 ChIP of the proximal limb tissue at embryonic day 11.5 after 4 hr of exposure to FGF8 and FGF10. Fold enrichment over nonspecific IgG recoveries and ±SEM from two independent experiments are plotted. DNA was quantified by q-PCR using the ZRS 3'LR oligos.

DOI: https://doi.org/10.7554/eLife.28590.010

The following figure supplement is available for figure 4:

**Figure supplement 1.** Analysis of 14Fp and limb bud cultures after treatment with the NIN and BJG inhibitors.

DOI: https://doi.org/10.7554/eLife.28590.011

demonstrated (*Kang et al., 2008*) that GABPα recruits the co-activator histone acetyltransferase (HAT) CBP/p300. Therefore, binding of GABPα to the ZRS and recruitment of p300 was examined in the 14Fp cell line (*Figure 5A*). ChIP-on-chip analysis on untreated cells for both GABPα and p300 showed an overlapping peak of enrichment for both factors, suggesting a co-occupancy over the ZRS. To further study the interaction between these factors, we performed co-immunoprecipitation experiments on nuclear extracts from 14Fp cells transfected with GABPα tagged with three copies of the flag epitope. Immunoprecipitation of endogenous p300 co-precipitates the flag-GABPα indicating an association between these two factors (*Figure 5B*). Western blot for p300 normalized against the histone H3 showed that p300 levels were not affected (*Figure 5—figure supplement 1B*). Based on the assumption that GABPα and p300 together have an important, yet undefined role in the activation of the regulatory element, we next addressed whether the presence of GABPα/p300 influenced ZRS activation. *Shh* expression was examined in cells induced with TSA after reduction of *Gabpa* expression. *Gabpα* small interfering RNA (siRNA) knockdown (*Figure 5C*) revealed decreased *Shh* expression after TSA treatment (*Figure 5D*). Conversely, overexpression of *Gabpα* using a doxycycline inducible vector leads to activation of *Shh* (*Figure 5E*, *Figure 5—figure supplement 1A*). In addition, ChIP analysis for H3K27ac in cells overexpressing GABPα showed an appreciable enrichment over the ZRS; whereas, the transcription factor ETV4, a repressor (see below) which restricts expression outside the ZPA, is displaced from the ZRS (*Figure 5F*). Under these conditions, enrichment of GABPα and p300 over the ZRS is also observed (*Figure 5G*). These data suggest that GABPα regulates *Shh* expression by modulating the acetylation status of H3K27 of the ZRS. Furthermore, FGF signalling plays a central role, since both GABPα and p300 are released from the ZRS when FGF signalling is inhibited by NIN and the enhancer is no longer poised (*Figure 4D–E*). FGF, therefore, mediates priming of the ZRS enabling the binding of GABPα which, in turn, recruits p300.

## ETV4 carries out its repressive role via interacting with HDAC2

ETV4/ETV5 binding represses *Shh* expression outside the ZPA in the limb bud (*Lettice et al., 2012*). Since there is a close association between GABPα and p300, we investigated the possibility that the repressive role of ETV4/ETV5 was related to HDAC activity. In order to investigate specific HDAC candidates, RNA-seq data obtained from the cell line and the distal and proximal portions of the limb bud showed nearly all the HDAC classes are represented in both 14Fp and in limb tissue between the cells and the tissue, with the exception of HDAC9 (*Figure 2—figure supplement 1B*). Most of the HDACs represented were subjected to ChIP in 14Fp (HDAC1, 2, 3, 4, 5, 6, 8, and 9) and HDAC2 appeared significantly enriched over the ZRS (data not shown) and is one of the most abundant *Hdac* in the RNA samples analysed (*Figure 2—figure supplement 1B*). To further examine the role of HDACs in ZRS activity, *Hdac2* and *Hdac1* expression levels were reduced by specific siRNAs in 14Fp cells (*Figure 6A*). Both are class I HDACs and often work in concert. *Hdac1* downregulation had no effect on *Shh* expression, while the reduction of *Hdac2* levels (by siRNA) in 14Fp showed an increase in *Shh* levels. Simultaneous reduction of both *Hdac1* and *Hdac2* did not have an additive

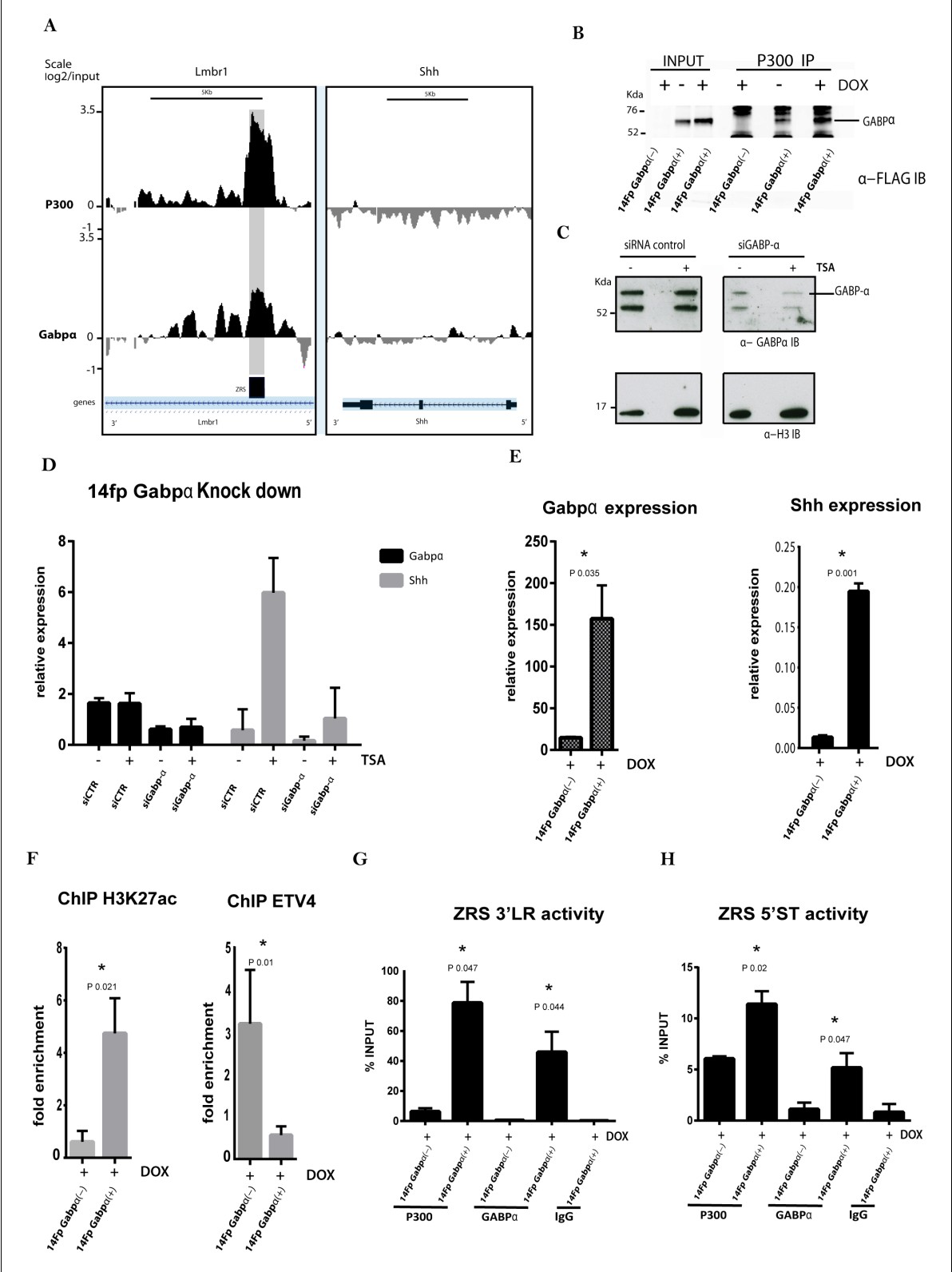

**Figure 5.** GABPα co-localizes with P300 and modulates ZRS acetylation status. (**A**) Chromatin immunoprecipitation (ChIP) analysis from two biological replicates on the 14Fp cell line using GABPα and p300 antibodies using tiling microarrays. Summary is presented using two different genomic regions, the y axis is log₂ for each ChIP/input DNA and the x axis represents a segment of DNA from the microarray. The DNA region containing the ZRS is highlighted by the grey shading. As controls, the whole of the *Shh* coding region plus promoter is shown. (**B**) 14Fp nuclear cell extracts from cells stably

*Figure 5 continued on next page*

*Figure 5 continued*

transfected with 3Xflag-Gabpα (*Figure 5—figure supplement 1E*) treated with or without doxycycline were analysed by immunoprecipitation with anti-p300 antibody followed by Western blot analysis with anti-flag-tag antibody. As control the empty vector plus doxycycline was used. (**C**) Western blot analysis with anti-GABPα of 14Fp nuclear cell extracts transiently transfected with Gabpα small interfering RNA (siRNA) (siGabp-α) or nonspecific siRNA (siCTR) and trichostatin A (TSA) treated. (**D**) Quantitative reverse transcriptase (qRT)-PCR was used to detect the messenger RNA (mRNA) levels of *Shh* (grey box) and *Gabpα* (black box) in 14Fp cells transfected with *Gabpα* siRNA or nonspecific siRNA. Eighteen hours after transfection, the cells were treated with 1 μM TSA for 24 hr. *Shh* and *Gabpα* levels were evaluated relative to control and normalized to glyceraldehyde 3-phosphate dehydrogenase levels from two biological replicates. (**E**) qRT-PCR to detect the mRNA levels of *Shh* (black box) and *Gabpα* (grey box) in 14Fp cells stably transfected with 3Xflag-Gabpα vector and an empty vector as control. Data points represent the mean ± SEM of three biological replicate. (**F**) Chromatin from 14Fp cells stably transfected with 3Xflag-*Gabpα* vector and an empty vector as control was analysed by ChIP for H3K27ac histone modification and ETV4 enrichment. DNA was quantified by q-PCR using the ZRS 5' spatiotemporal (5'ST) activity oligo set. Data are represented as mean ± SEM of the fold enrichment over nonspecific IgG recoveries from two independent experiments. (**G–H**) Chromatin from 14Fp cells stably transfected with 3Xflag-*Gabpα* vector and an empty vector as control was analysed by ChIP for GABPα and P300. DNA was quantified by q-PCR using the ZRS 3' long-range (3'LR) and 5'ST activity oligos sets. Average of percentage of input ±SEM from two independent experiments are plotted.
DOI: https://doi.org/10.7554/eLife.28590.012

The following figure supplement is available for figure 5:

**Figure supplement 1.** Analysis of 14Fp cells after mis-expression of of flag-tagged GABPa.
DOI: https://doi.org/10.7554/eLife.28590.013

effect on the induction of *Shh* (*Figure 6A*). To investigate the relationship between the repressor activity of the ETV genes and HDAC2, we performed ChIP-on-chip for these two factors to visualize their distribution over the ZRS (*Figure 6B*). Since in the 14Fp cell line ETV5 expression is low (approximately eightfold to tenfold lower than ETV4) (*Figure 2—figure supplement 1A–B*), depletion of ETV5 activity was not necessary in this analysis. ETV4 exhibits two peaks encompassing the ZRS which correspond to the two ETV4 binding sites (*Lettice et al., 2012*); interestingly, the HDAC2 peak overlapped one of these peaks located at the 3' end which encodes the long-range activity of the regulatory element (*Lettice et al., 2014*). We next tested if HDAC2 and ETV4 were able to physically interact; analysis showed that endogenous ETV4 co-immunoprecipitated with HDAC2 (*Figure 6C*). In addition, the negative role of ETV4 on *Shh* expression in 14Fp cells (*Figure 6D*) was investigated. *Etv4* levels were reduced with siRNA, resulting in a ~40% decrease in *ETV4* and a significant activation of *Shh* expression (*Figure 6D*). ChIP using anti-ETV4 and HDAC2 antibodies performed on TSA treated cells showed that ETV4 together with HDAC2 are displaced from the ZRS (*Figure 6E*), suggesting that ETV4 opposes GABPα activity by maintaining lower levels of H3K27ac.

## Discussion

### The distal limb bud is poised for expression

The ZPA is the organizing centre of the early developing limb bud and the restricted expression of *Shh* at this location along the posterior boundary is crucial for correct specification of digit identity and number. Various regulatory inputs are essential to acquire this spatial specific pattern of expression (*Figure 7*). We showed that an initial input is the event that primes the ZRS, such that the enhancer is poised but transcriptionally inactive. The ZRS priming occurs in a broad region of the distal limb bud mesenchyme that includes tissue that will not express *Shh* in addition to the ZPA. Furthermore, we undertook studies to identify the signalling pathway involved in the induction of ZRS priming. Distal limb mesenchyme, referred to as the progress zone, is known to be under the influence of the FGFs produced in the AER (*Laufer et al., 1994*; *Niswander et al., 1994*; *Crossley et al., 1996*; *Vogel et al., 1996*; *Ohuchi et al., 1997*) and we showed that FGFs can induce ZRS priming in distal mesenchyme; whereas, inhibition of FGF signalling results in chromatin changes and loss of H3K4me1 and loss of transcription factor binding suggesting that the ZRS is no longer recognized as a poised enhancer and is in a 'closed' configuration. Thus, one role of FGF signalling is the establishment and maintenance of ZRS priming. Activation of the ZRS, therefore, appears to be a two-step process; ZRS priming occurring broadly in the distal mesenchyme which is a prerequisite for subsequent action by other signals in the posterior region containing the ZPA to activate the ZRS.

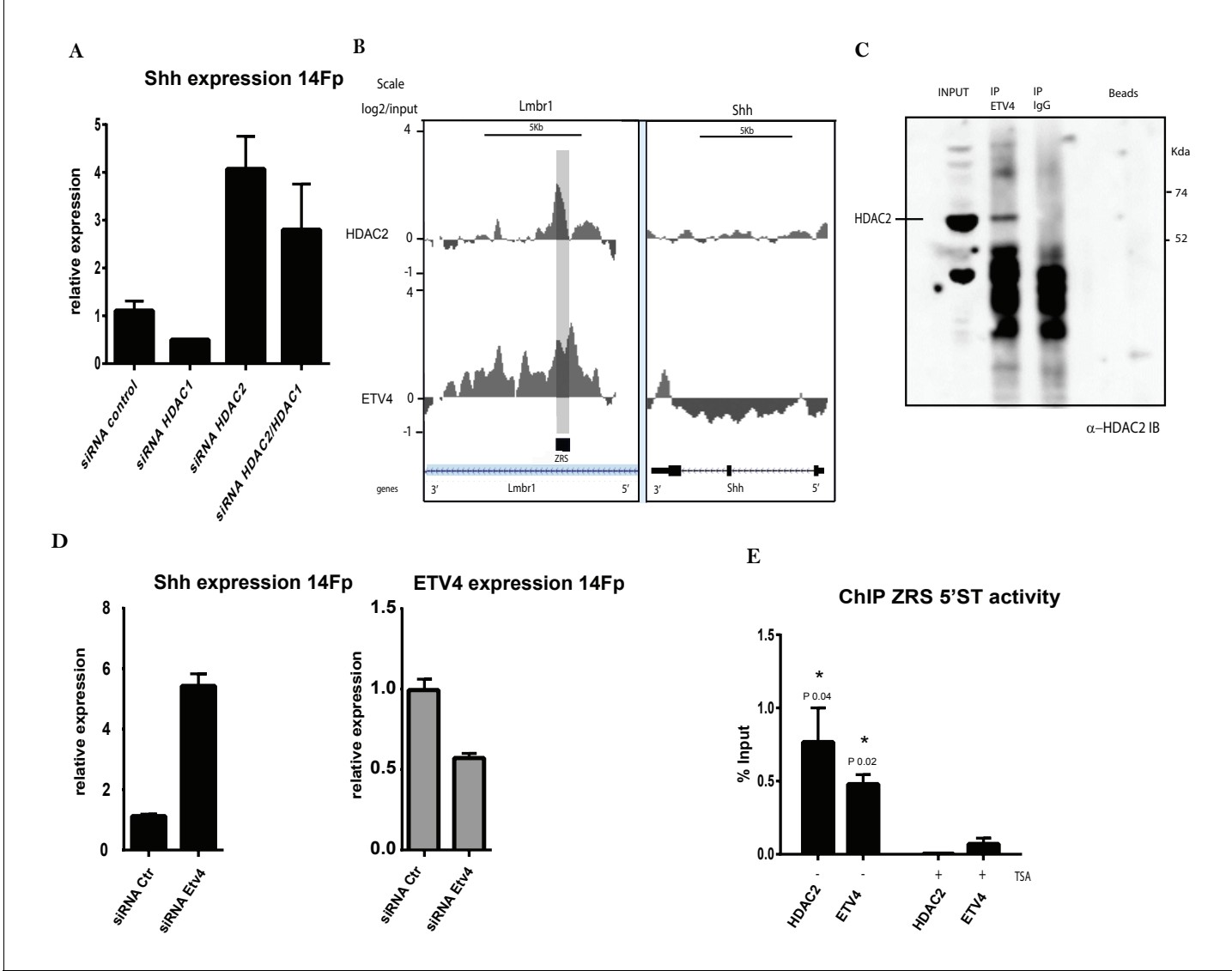

**Figure 6.** ETV4 acts as a repressor via interactions with HDAC2. (**A**) Quantitative reverse transcriptase (q RT)-PCR to detect the messenger RNA (mRNA) levels of *Shh* (black box) in 14Fp cells transfected with HDAC1 and HDAC2 small interfering RNA (siRNA) either alone or combined and with nonspecific siRNA as control. Data were collected after 18 hr of transfection. *Shh* levels were evaluated relative to control and normalized to glyceraldehyde 3-phosphate dehydrogenase levels. Data points represent the average of triplicate determinations ± SEM. (**B**) ChIP from two biological replicates using the 14Fp cell line and anti-ETV4 and HDAC2 antibodies analysed by hybridizing to tiling microarrays (*Figure 5—figure supplement 1B*). Summary is presented using two different genomic regions, the y axis is $\log_2$ for each ChIP/input DNA and the x axis represents a segment of DNA from the microarray. The DNA region containing the ZRS is highlighted by the grey shading. As controls, the whole of the *Shh* coding region plus promoter is shown. (**C**) 14Fp nuclear cell extracts were analysed by immunoprecipitation with anti-ETV4 and IgG antibodies followed by Western blot analysis with anti-HDAC2. (**D**) qRT-PCR to detect the mRNA levels of *Shh* (black box) and *Etv4* (grey box) in 14Fp cells transiently transfected with ETV4 siRNA (siRNA Etv4) or nonspecific siRNA (siRNA Ctr). Data points represent the average of triplicate determinations ± SEM. (**E**) Shown are results from chromatin immunoprecipitation (ChIP) analysis using anti-HDAC2 and ETV4 antibody after 24 hr of trichostatin A (TSA) treatment. Recovered DNA sequences were quantified by quantitative PCR using 5' spatiotemporal (5'ST) oligo set. Average percentage of input and ±SEM from two independent experiments are plotted. The IgG did not give detectable signal.

DOI: https://doi.org/10.7554/eLife.28590.014

The poised state of the enhancer is notable in light of the response of the ZRS to the point mutations that cause PPD2 and other associated skeletal abnormalities (*Anderson et al., 2012*). These mutations cause misregulation of *Shh* expression in the developing limb bud such that *Shh* expression occurs at an ectopic anterior site in the distal mesenchyme in addition to the ZPA. Ectopic

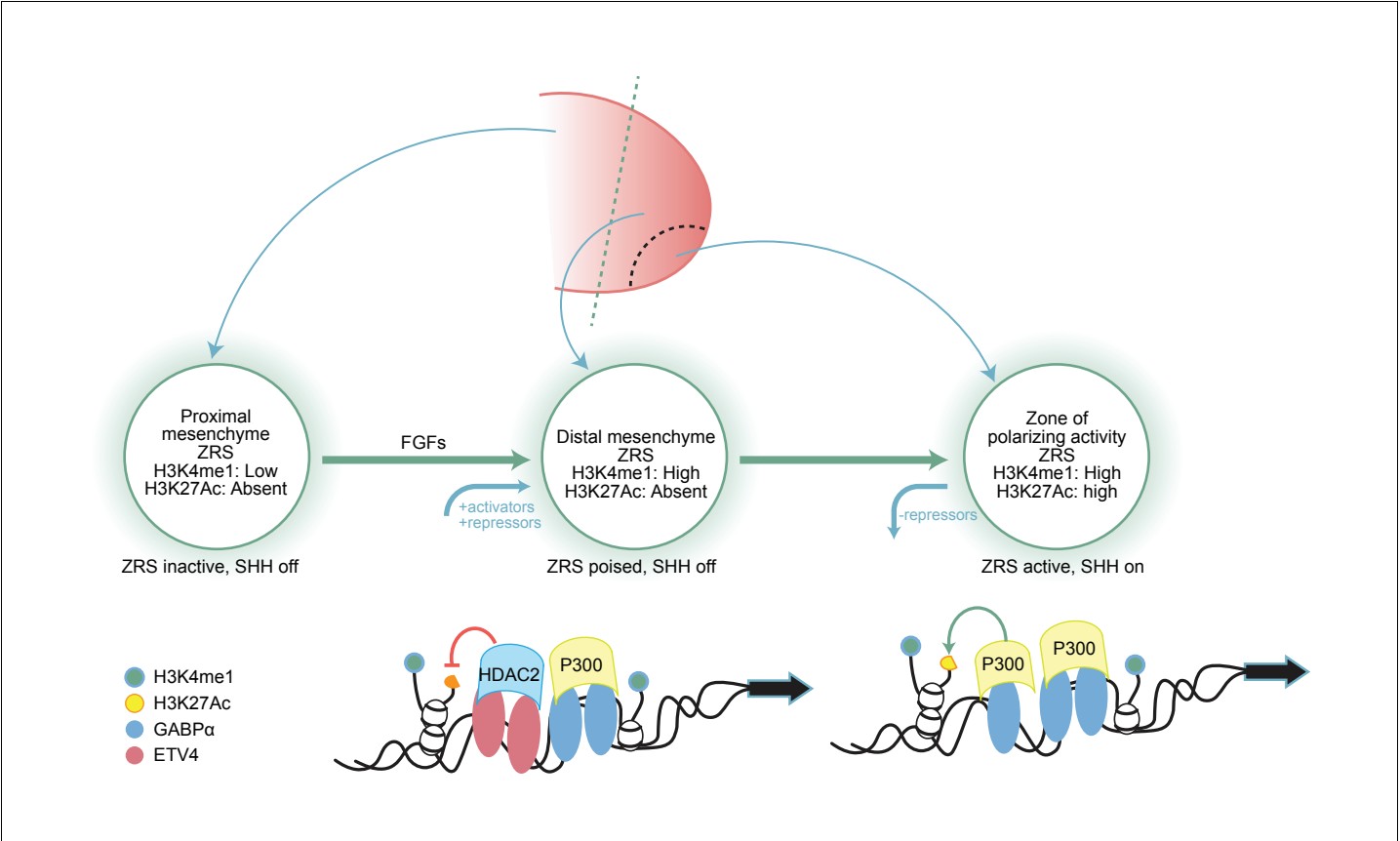

**Figure 7.** Fibroblast growth factor (FGF) signalling is responsible for priming the ZRS for local chromatin changes mediated by E26 transformation-specific factors. Summary model showing how FGF signalling in the distal mesenchyme regulates the ZRS poised state and that this allows the stepwise recruitment of transcriptional regulators to the ZRS. FGF signalling has a dual fundamental role; firstly, as an activator it is responsible for priming and maintaining the ZRS and secondly, as a repressor inducing the expression on ETV4 which restricts the expression of the ZRS. ETV4 (red oval) interacts with HDAC2 (light blue rectangle) to maintain the inactivity of the poised ZRS enhancer throughout the distal mesenchyme, while GABPα (blue oval) acts within the ZPA domain to recruit p300 (yellow rectangle), trigger H3K27 (orange circle) acetylation, and thereby activate *Shh* transcription.
DOI: https://doi.org/10.7554/eLife.28590.015

expression of all ZRS mutations examined, thus far, is restricted to the distal mesenchyme suggesting that there is a requirement for the ZRS to be in a poised state in order to be activated and demonstrates a mechanism for *Shh* ectopic activation restricted to the distal mesenchyme of the limb bud.

## Levels of GABPα and ETV4 regulate *Shh* expression

Attempts to understand how developmental enhancers operate over such large distances have led to the general idea that chromatin looping assists the interactions of a long-range enhancer with its target promoter (*Sanyal et al., 2012*). In the limb bud, 3C assay revealed a physical interaction between the ZRS and the promoter (*Amano et al., 2009*). Here, we confirmed, that upon activation of *Shh* expression in the limb-derived cell line, there is highly specific co-localization of the ZRS with the *Shh* promoter fragment. Thus, the *Shh* limb regulator in the cell line exists in two states; one in which the ZRS is poised reflecting its origin from the distal mesenchyme of the developing limb bud and secondly, in an active state in which the ZRS and the promoter recruit Pol II and interact. Hence, the cell line reveals the chromatin dynamics that occur during long-range activation.

GABPα/ETS1 and ETV4/ETV5 have antagonistic effects on *Shh* activity in the limb. Occupancy at multiple GABPα/ETS1 sites in the ZRS regulates the position of the ZPA boundary, whereas ETV4/ETV5 binding acts as repressors to restrict expression outside the ZPA. We have hypothesized that the balance between these activator and repressor factors is responsible for regulating activity levels

in the developing limb bud (*Lettice et al., 2012*). Here, we showed that GABPα and ETV4 bind to the ZRS in the cell lines similar to that in the limb bud. GABPα is associated with p300 HAT at the ZRS in the cell lines. Previous reports demonstrated that GABPα physically interacts with p300 in myeloid cells (*Resendes and Rosmarin, 2006*). Activation of AChR (nicotinic acetylcholine receptor) gene in subsynaptic nuclei in adult skeletal muscle is mediated by GABPα, which recruits the HAT p300 to the AChR ε-subunit promoter (*Ravel-Chapuis et al., 2007*). Furthermore, the surface of the GABPα OST domain binds to the CH1 and CH3 domains of the co-activator HAT CBP/p300 (*Kang et al., 2008*). Here, we showed in the limb-derived cells that a similar interaction of these two factors occurs. In addition, overexpression of GABPα results in an increase in the levels of H3K27ac at the ZRS. A correlation between *Gabpα* expression and the presence of ETV4 at the ZRS is also observed; in that, expression of *Gabpα* results in a displacement of ETV4 at the enhancer. Thus GABPα operates at the ZRS to increase levels of H3K27ac by recruiting p300 and decreasing levels of the repressor ETV4.

ETV4 appears to operate in the opposite manner and is one of the dominant factors in maintaining the ZRS in a poised state. Reduction of ETV4 levels is sufficient to activate *Shh* expression. ETV4 interacts with HDAC2 and we further showed the co-localization of HDAC2 and ETV4 at the ZRS in the cell line. Similar to the downregulation of ETV4 the knockdown of HDAC2 induces *Shh*. Activation of *Shh* by TSA releases ETV4 and HDAC2 from the ZRS. Hence activation of the ZRS by both the overexpression of GABPα and treatment with TSA is associated with loss of ETV4 binding which acts directly on HDAC activity.

The FGFs play a central role in *Shh* expression by ensuring that the ZRS maintains its primed state throughout the distal limb mesenchyme. This widespread priming, however, opens the ZRS for potential ectopic activation as occurs in preaxial polydactyly and associated phenotypes (*Anderson et al., 2012*). ETV4 is a repressor that restricts this activity and FGF signalling also induces the levels of ETV4 in the distal mesenchyme of the limb bud (*Mao et al., 2009*; *Zhang et al., 2009*). The FGFs, therefore, operate by regulating two contrasting events; firstly, by effecting the chromatin in the distal mesenchyme ensuring that the ZRS is maintained in a poised state and secondly, acting counter to enhancer activation by inducing the level of a repressor that ensures *Shh* expression does not occur outside the region of the ZPA.

## Materials and methods

### Cell lines, Transfections, and Treatments

The 14Fp cell line was derived from the posterior third of distal forelimb buds from an Immortomouse (H-2k-tsA58) (*Jat et al., 1991*). Cells are plated in DMEM (Invitrogen, Loughborough, UK) with 10% Foetal Calf Serum (Sigma-Aldrich, Gillingham, UK), Penicillin/Streptomycin and 20 ng/ml Interferon (Peprotech, London, UK). Cells are grown at 33°C the permissive temperature for the temperature-sensitive T antigen and were passaged as necessary but no later than passage 12. Cells biological replicates are intended as cells cultured separately and treated/analysed at a different passage. Cellular identity was confirmed by RNA expression analysis of specific genes and immortomouse markers and verified to be mycoplasma free. Knockdown of endogenous proteins was performed in 14fp cells after siRNA transfection using Dharmafect one solution (Dharmacon). Briefly, cells were seeded in six-well plates to 40% confluence and after 24 hr were transfected using 25 nM of each siRNA pool and 10 μl of the transfection reagent. The transfection medium was replaced after 12 hr and cells were grown for another 12 hr. Cells were collected 24 hr after the transfection for analyses. siRNA pools were purchased from Thermo Fisher Scientific (Ambion): Gabpα (s66354, s66355), Etv4 (s71463, s201776), Hdac2 (s67417, s67416), Hdac1 (s119557, s119558), and nontargeting siRNAs (control) (D-001810-02). The siRNA were used separately. Unless otherwise specified, the cells were treated with HDAC inhibitors TSA (1 μM) for 24 hr before cell harvest and NIN (100 nM) and BGJ (2.5 μM) for 4 hr. Cell cultures were incubated, when necessary, with 500 ng/ml of FGF10 (R&D Systems, Abingdon, UK catalogue no. 345-FG) and 100 ng/ml of FGF8 (Abcam, Cambridge, UK, ab205522) combination for 6 hr after NIN treatments.

## Mouse limb cultures

Limb buds were dissected from E11.5 mouse embryos and cultured as described by *Havis et al. (2014)* with the following modifications. Limb dissections obtained from different embryos were considered as biological replicates. Briefly distal dissections were treated with NIN (100 nM) for 4 hr. Proximal dissections were incubated with 500 ng/ml of FGF10 (R and D Systems, catalogue no. 345-FG) and 100 ng/ml of FGF8 (ABCAM, ab205522) combination for 6 hr. Inhibitors were diluted in DMSO. Media with buffers only were used as controls. After treatments, explants were processed for qRT-PCR or ChIP.

Trypan blue staining was performed on limb dissections by following a reported protocol with slight modifications (*Siddique, 2012*). Total limb dissections were collected and treated 4 hr with NIN or media alone. The limbs were then transferred to trypan blue stain and kept in shaking condition for 20 min. Eight limb dissections were directly stained after collection (T0). The samples were washed thoroughly with PBS solution again for 15 more minutes and observed by microscope and imaged to check for any cell damage.

## ChIP and antibodies

Crosslinked ChIP was performed as described (*Stock et al., 2007*) from approximately $10^7$ cells per experiment. All antibodies used in this study have been previously reported as suitable for ChIP and/or ChIP-seq, p300 (sc-585, Santa Cruz Biotechnology, Heidelberg, Germany), H3K4me1 (ab8895, Abcam), H3K27ac (s39133, Active Motif, La Hulpe, Belguim), HDAC2 (ab16032), GABPα (sc-22810, Santa Cruz Biotechnology), with the exception at ETV4 (ABE635, Millipore, Loughborough, UK). All statistical analyses were performed using a two-tailed Student's *t*-test.

## 3C and 4C

3C was conducted according to the protocol described by *Stadhouders et al., 2013* with minor alterations. In brief cells were treated with 1 mM TSA for 18 and 24 hr or DMSO as a control and fixed in 2% formaldehyde solution for 10 min. Glycine (0.125 M) was used to quench the reaction. After a PBS wash and 15 min incubation in lysis buffer, the solution was spun down and nuclei stored at −80°C until needed. If more than $5 \times 10^5$ cells were used, pellets were made up in 1.2× restriction buffer and divided into four aliquots to reduce formation of aggregates. Primary restriction enzyme digestion was conducted using 800 U HindIII-HF restriction enzyme (NEB) at 37°C on each aliquot. Before ligation aliquots were combined and T4 DNA ligase (New England Biolabs, Evry, France) added and incubated at 16°C overnight. HindIII digested samples were analysed on a 0.6% agarose gel and appear as high molecular weight smear running from roughly 4 to 12 kb (*Figure 2—figure supplement 3D*, *Figure 3—figure supplement 1A*). 3C libraries were analysed on 0.6% agarose gel and appear as a high molecular weight band (around 12 kb) (*Figure 2—figure supplement 3E*, *Figure 3—figure supplement 1B*). To this point the 3C libraries were analysed (see below). In order to make 4C libraries a second restriction digest was performed using the four-cutter MluCI (Roche, Burgess Hill, UK) at 37°C overnight and second ligation with T4 DNA ligase (New England Biolabs). MluCI digested samples were analysed on a 1.5% gel appearing as smear between roughly 0.3 and 1 kb (*Figure 2—figure supplement 3F*). Finally, 4C libraries were purified using QIA quick PCR purification kit (Qiagen, Germantown, MD, USA) to produce final purified 4C libraries. After PCR amplification and purification (Table ST2), sequencing adaptors were ligated and 4C libraries sequenced using in-house Ion Proton sequencing.

## Analysis of 4C libraries

De-multiplexed sequencing reads (fastq files) can be summarized: first we trimmed known bait sequence using cutadapt (*Martin, 2011*) and selected only those reads where known viewpoint-associated sequence was present. Reads were mapped to the mouse reference genome (build mm9) using bowtie2 (*Langmead and Salzberg, 2012*) with the very-sensitive flag 3. Alignments were filtered with a MAPping Quality (MAPQ) score >30 to select for high-confidence alignments using SAMtools (*Li et al., 2009*). Contacts were then normalized using the r3cseq R package and assigned false discovery rate q-values to interactions, with the aim of finding those significantly over-

represented relative to expectation. The normalization procedure for 4C data is adapted from a previous method for normalizing deepCAGE data between samples (*Balwierz et al., 2009*).

## Analysis of 3C libraries

Digestion efficiency and sample purity was assessed as described previously (*Hagège et al., 2007*). Primers were designed using Primer3 with an anchor primer in the fragment at the 5' end of the ZRS and in potential interacting fragments around the Shh promoter, gene body, 3' end, and gene desert. qPCR was carried out using the Roche LightCycler 480 SYBR Green I Master and Roche LightCycler 480 probe Master on a Roche LC480 according to the instructions of the manufacturer (Roche). Two PACs, RCPI21-542n10 (148 kb long covering Rnf32 to 5' of LMBR1) and RCPI21-508F15 (203 kb long covering Shh and 150 kb 5') obtained from RPCI21 library (HGMP Resource Centre, Cambridge, United Kingdom), were used as a PCR control template. The PAC clones were cut with *Hin*dIII and equimolar amounts re-ligated by T4 DNA ligase. All primer pairs were tested on a standard curve of the BAC control library and yielded PCR efficiencies > 1.7. The presence of a single PCR product was confirmed and melting curves analysed. Cycling conditions were: 95°C for 5 min, 40 cycles of 95°C for 15 s, 60°C for 30 s, 72°C for 30 s . qPCR data were normalized to ZRS 3'LR ChIP oligos as a loading control. 3C oligos from *Splinter et al. (2006)* were used to control for interaction frequencies between samples. ZRS 3'LR ChIP oligos cycling conditions were 95°C for 5 min, 40 cycles of 95°C for 10 s, 60°C for 30 s, 72°C for 30 s. The cycling conditions for the interacting fragments and the anchor were 95°C for 5 min, 45 cycles of 95°C for 15 s, 60°C for 1 min, 72°C for 1 s. Data analysis was carried out according to *Hagège et al. (2007)* and is presented as relative crosslinking frequency. The primers used for the chromatin conformation capture interaction studies are listed in Table S2.

## ChIP and tiling microarrays

Cells from dissected E11.5 limbs and 14fp were fixed with 1% formaldehyde (25°C, 10 min) and stopped with 0.125 M glycine. Crosslinked ChIP was performed as described (*Stock et al., 2007*). In brief, the nuclei were sonicated using a Diagenode Bioruptor (Leige, full power 30 s on, 30 s off, in an ice bath for 50 min) to produce fragments of <300 bp. Chromatin (350 mg) was incubated with 5 mg prebound (to Protein A or G magnetic beads, Invitrogen) IgG (Santa Cruz, sc-2025) or the previously mentioned antibodies, raised to in the presence of 50 mg of BSA, washed, and eluted. Reverse crosslinked DNA was purified with Proteinase K (ThermoFisher Scientific) and QIAGEN PCR purification kit. qPCR of the ChIP experiments was carried out using equal concentrations of input, IgG, and Chip DNA using a Sybr Green (Roche) reaction. From each biological replicate three technical replicates were analysed. Enrichment values for ChIP samples from limb bud sections extracts are presented either as percentage of input or as fold differences relative to IgG and normalized to input with the formula $2([CtIgG-CtInput]-[CtAb-CtInput])$ where Ct values are threshold cycles. All biological replicates were carried out in duplicate unless stated. Primers used for ZRS and the controls are shown in Table ST1. For the custom Nimblegen tiling arrays ChIP DNA and input DNA ChIP DNA and input DNA were amplified (WGA2 kit, Sigma), labelled, and hybridized according to the manufacturer's protocol to a 354,999 unique probe custom microarray containing specific tiled regions of the mouse genome (Nimblegen, Roche). For the Agilent arrays a custom tiling array was designed including some of the genes involved in limb development, including gene deserts associated with such genes. GEO accession number for the ChIP data is GSE104074 and GSE104208.

## Nimblegen arrays

Microarray data were analysed in R/Bioconductor (http://genomebiology.com/2004/5/10/R80) with the Epigenome (PROT43) protocol (https://www.epigenesys.eu/images/stories/protocols/pdf/20111025114444_p43.pdf) with the following parameters. The mean signal intensity of the two replicate probes on each array was taken. Loess normalization was used within arrays to correct for the dye bias, and scale normalization was used within the replicates group to control interarray variability using the R package (*Ritchie et al., 2015*). The $log_2$ enrichment for each ChIP group was calculated by subtracting the mean of $log_2$ input intensities from the mean of d enriched intensities and averaging over the two biological replicates.

## Agilent arrays

The Median Signal was extracted from the scanned image files and processed using the R package (*Toedling et al., 2007*). Probe intensities were transformed from raw values into background-corrected normalized log ratios (ChIP/Input) using Ringo's preprocess method (VSN normalization). Smoothing over individual probe intensities was performed using a sliding window of 1000 bp along the chromosome and replacing the intensity at the genomic position by the median over the intensities of those reporters inside the window.

## Immunoprecipitation of FLAG fusion proteins

The pSV40-Tet3G- pTRE3G-mCherry- Gabpα plasmid was generated using the In-Fusion cloning technology (Clontech, St Germaine en Laye, France catalogue no. 639649). First, the pTRE3G-mCherry vector (Clontech, catalogue nos. 631160 and 631175) was linearized using a unique restriction site between the origin of replication and the Tet-responsive promoter (pTRE3G). Second, the sequences encoding for SV40 promoter, Kanamycin/Neomycin resistance, internal ribosome entry site IRES2, and Tet-ON 3G transactivator were inserted. Third, we removed the Ampicillin resistance cassette. Lastly, the sequence of the self-cleaving peptide P2A was cloned between the mCherry fluorescent marker and the 3xFLAG tagged Gabpα gene, in order to generate a bicistronic expression under the control of the TRE3G promoter and an empty control vector was also used. Overexpression analyses were performed in 14fp cells by transfecting plasmids using Lipofectamine LTX with plus reagent (ThermoFisher Scientific) and following standard manufacturer's protocol. Nuclear extracts used in the immunoprecipitation were prepared from 14fp using a NE-PER nuclear and cytoplasmic extraction kit (ThermoFisher Scientific). The nuclear extract was incubated with anti-FLAG M2 affinity gel (Sigma) for 3 hr at 4°C. The beads were washed three times with washing buffer. The immunoprecipitates were eluted with 1× SDS buffer, separated on a 4–20% Novex Tris-Glycine SDS-PAGE gel (Invitrogen), transferred to a PVDF membrane (Millipore), incubated with anti-FLAG (Sigma) or with anti-GABPα (Santa Cruz, sc-22810).

## Gene expression analysis, RNA library Preparation, Sequencing, and Analysis

Total RNA was prepared using Trizol reagent (Invitrogen) according to manufacturer's protocol (for limb buds, dissected anterior, posterior, distal, and proximal tissue was dissociated into single cell suspensions in Trizol using a syringe fitted with a 25G (0.5 mm) needle (Sigma Aldrich, BD Microlance), followed by acid phenol:chloroform:isopropyl alcohol extraction, and then digested with 2U DNaseI (ThermoFisher Scientific, Ambion) for 30 min at 37°C). RNA was reverse-transcribed to complementary DNA (cDNA) using QuantiTect Reverse Transcription Kit (Qiagen). The quantitative real-time PCRs were performed in a 7300 system (ThermoFisher Scientific, Applied Biosystems, Life Technologies) by using LightCycler 480 SYBR Green I Master (Roche) and gene specific primer sets for shh, Gabpα, and Etv4. The cycle threshold (CT) values from all quantitative real-time PCR experiments were analysed using ΔΔCT method. Data were normalized to GAPDH and expressed as fold changes over that in control treatment group. From each biological replicate three technical replicates were analysed. All statistical analyses were performed using a two-tailed Student's *t*-test.

RNA sequencing was conducted by GATC Biotech (Konstantz, Germany). Samples were only submitted with an OD 260/280 ratio ≥1.8, a 260/230 ratio ≥1.7, and a RNA Integrity Number value ≥8 as detected by Agilent Technologies 2100 Bionalayser. The InView Transcriptome Explore service provided by GATC was used to provide a randomly primed and amplified cDNA library with Illumina adaptors ready for sequencing. Illumina sequencing was conducted producing 50 bp single end reads and a guarantee of over 30 million reads per sample. All samples were analysed on the main Galaxy server by first checking sequence quality by FastQC. Reads were then trimmed and any Illumina sequencing adaptors removed as appropriate and aligned to the mouse genome (mm9, NCBI 37) using Tophat2 (Galaxy Tool Version 0.9). The results for each condition were fed into Cuffdiff (Galaxy Tool Version 2.2.1.3) and visualized using the R Bioconductor package CummeRbund (Release 3.2). This process was repeated using RNA isolated from both immortalized limb cell lines and isolated limb tissue. Two biological replicates were analysed for each condition.

## Acknowledgements

We thank the staff at the Evans Building and especially Anna Thornburn for expert technical assistance. We also thank Prof. N Hastie and Prof. W Bickmore for critically reading the manuscript. This work was supported by an MRC core grant. We also thank Dr. Laura Lettice who kindly provided us the PAC vectors and the 3C libraries from anterior and posterior limb tissue.

## Additional information

### Funding

| Funder | Author |
| --- | --- |
| Medical Research Council | Silvia Peluso<br>Adam Douglas<br>Alison Hill<br>Carlo De Angelis<br>Benjamin L Moore<br>Graeme Grimes<br>Giulia Petrovich<br>Abdelkader Essafi<br>Robert E Hill |

The funders had no role in study design, data collection and interpretation, or the decision to submit the work for publication.

### Author contributions

Silvia Peluso, Conceptualization, Data curation, Formal analysis, Validation, Investigation, Visualization, Methodology, Writing—original draft, Project administration, Writing—review and editing, Analysed and interpreted the data and prepared the manuscript. designed and performed the majority of experiments; Adam Douglas, Data curation, Formal analysis, Validation, Methodology, performed the 4C experiment; Alison Hill, Resources, Validation, Methodology, Contributed to production of the experimental data; Carlo De Angelis, Resources, Validation, Methodology, contributed to production of the experimental data; Benjamin L Moore, Software, Visualization, Writing—review and editing, analysed the 4C experiment; Graeme Grimes, Software, Validation, Writing—review and editing, Analysed the the ChiP-chip data; Giulia Petrovich, Validation, Methodology, Writing—review and editing, Generated the 3X flag vector; Abdelkader Essafi, Conceptualization, Writing—review and editing, provided scientific support; Robert E Hill, Conceptualization, Supervision, Funding acquisition, Investigation, Writing—original draft, Project administration, Writing—review and editing, analysed and interpreted the data, prepared the manuscript and supervised the study

### Author ORCIDs

Silvia Peluso  http://orcid.org/0000-0003-1803-5300
Benjamin L Moore  http://orcid.org/0000-0002-4074-1933
Robert E Hill  http://orcid.org/0000-0003-2848-1080

### Ethics

Animal experimentation: Experiments using mice were conducted in accordance with the UK Animals (Scientific Procedures) Act 1986, with appropriate personal and project licences in place.

### Decision letter and Author response

Decision letter https://doi.org/10.7554/eLife.28590.018
Author response https://doi.org/10.7554/eLife.28590.019

## Additional files

**Supplementary files**
• Supplementary file 1. (A) List of oligos used for qRT-PCR, q-PCR and ChIP. (B) List of oligos used for 3C and 4C analysis.
DOI: https://doi.org/10.7554/eLife.28590.016

• Transparent reporting form
DOI: https://doi.org/10.7554/eLife.28590.017

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
