## [Decision Letter]

Thank you for submitting your article "FGF primes the limb specific ZRS enhancer for chromatin changes that balance histone acetylation mediated by ETS factors" for consideration by *eLife*. Your article has been favorably evaluated by Didier Stainier (Senior Editor) and three reviewers, one of whom, Lee Niswander (Reviewer #1), is a member of our Board of Reviewing Editors. The following individual involved in review of your submission has agreed to reveal their identity: Deneen Wellik (Reviewer #2).

The reviewers have discussed the reviews with one another and the Reviewing Editor has drafted this decision to help you prepare a revised submission.

Summary:

The manuscript by Peluso, Hill and colleagues focuses on the ZRS enhancer that lies 800Kb from the *Shh* gene and regulates *Shh* in the developing limb. This long-range enhancer and the wealth of human and mouse genetic data and molecular mechanisms elucidated to date on activators and repressors make for an excellent system to study enhancer/promoter interactions in the context of a complex mammalian tissue. Here the Hill group provide significant insights into how the balance of opposing chromatin modifying enzymes on the ZRS are involved in the epigenetic regulation of the ZRS enhancer and how developmental regulators of *Shh* expression affect the chromatin state of the ZRS. A strength of the paper is that the combined evidence provides a coherent, holistic model of how activation localized to the ZPA of the limb bud occurs. The existence of a poised, then activated state is not highly novel and the technology and chromatin marks analyzed are standard but the biological insights into how a developmental enhancer is deployed in a spatiotemporal and long-range manner are excellent.

This work shows clear lines of evidence that the ZRS *Shh* enhancer is poised in the distal, but not proximal limb bud as judged by H3K4me1. The H3K27ac, a mark of active enhancers, is shown to be higher in posterior, but not the anterior limb bud. While expected factors do not lead to *Shh* activation in a cell line derived from posterior-distal limbs, tricostatin A leads to *Shh* expression and this correlates with H3K27ac. FGF inhibition leads to loss of the primed or poised state, and correlations with GABPalpha binding and activation and *Etv4* repression are made (confirming previous data from this and other labs). A model is presented in which the authors propose that *Shh* expression in ZPA requires first that the enhancer become poised, which requires FGF signaling, and then activated.

Essential revisions:

Although there is enthusiasm, the reviewers raised the following criticisms.

Use of the cell line appears to have led to exploration of the epigenetic states, and as such, contributes to the story, but the true integrity of this cell line as a bona fide system for distal limb bud mesenchyme is not completely compelling based on the expression comparisons in supplemental data. No mention of the cell line is found in Abstract, for instance. Further comparisons of this cell line will be required to support this and confirmation and conclusions regarding the integrity of this cell line and its broad usefulness as a distal, posterior limb population should not be concluded here. This reviewer would recommend that there is less emphasis on this aspect in Results and Discussion. How reflective of posterior, distal limb bud this cell line is more broadly could be explored in a future publication.

In Figure 1, the fold enrichment of H3K27ac at the ZRS is not so strong (ca. 2 fold) in the limb bud that contains ZPA, but much stronger enrichment is seen in control of the cultured cells (more than 10 fold in Figure 2). How is this difference explained? It would be thought that acetylation of H3K27 at the ZRS in ZPA is abolished under the culture condition due to detachment of AER signals. Is it not the case? Direct comparison of ChIP-qPCR for H3K27ac between in vivo and the culture condition may be one way to address this question. Alternatively, a time course of deacetylation at H3K27 in ZRS in cultured limb bud cells should be presented.

Procedure of the 4C-seq is not clearly stated, and therefore the present result is not scientifically sound. The anchor (bait) point is not clearly shown. Digestion efficiency with restriction enzymes is not evaluated. Method of the 4C-DNA mapping is not well described. Mapping rate per total read counts is also lacking. Another concern is weak and very specific interaction between the ZRS and *Shh*. In a previous report from the same group (Anderson et al., 2014), it was stated that the *Shh* locus forms a topological domain where various genomic regions frequently interact with each other, consistent with Hi-C data from other groups. In Figure 3, ZRS seems to exclusively interact with *Shh*. It is an intriguing possibility that the TSA treatment facilitate a specific interaction in cultured cells. If it is the case, reproducibility of such weak interaction should be shown by experimental replicates of the 4C-analysis. Alternatively, 3C-qPCR could be fine to evaluate such specific interactions.

Figure 4 and legend, it is unclear if a specific region is analyzed (the ZRS? 5', 3'?). As presented it appears that H3K4me1 ChIP is severely altered at all loci by FGF inhibition with nintedanib. Presumably this is data from a single locus but it does lead to the question as to whether the tissue is severely compromised/dying. The relative expression of *Grem1* goes up only modestly and relying on proximal tissue results is a biased comparison of tissue health since these cells are more mature.

*Etv4* null mutant mouse is viable, and exhibits no apparent defect in the limb morphology (Laing et al., 2000). This is inconsistent with the claim of this study that binding of ETV4 to the ZRS suppresses *Shh*. Is *Etv4* indispensable for suppression of *Shh* in the fp14 cells, whereas in general ETV4 and other ETV proteins including ETV5 are functionally redundant. At a minimum the authors need to include a discussion about the possibility that other ETV proteins suppress *Shh* expression. It is preferable that the expression level of *Etv5* in the cell culture conditions is analyzed.

There is an opportunity to better summarize and highlight the contribution of this study in the conclusions; it currently reads more as a summary of results, both current and past, and could be shortened.

---

## [Author Response]

Essential revisions:Although there is enthusiasm, the reviewers raised the following criticisms.Use of the cell line appears to have led to exploration of the epigenetic states, and as such, contributes to the story, but the true integrity of this cell line as a bona fide system for distal limb bud mesenchyme is not completely compelling based on the expression comparisons in supplemental data. No mention of the cell line is found in Abstract, for instance. Further comparisons of this cell line will be required to support this and confirmation and conclusions regarding the integrity of this cell line and its broad usefulness as a distal, posterior limb population should not be concluded here. This reviewer would recommend that there is less emphasis on this aspect in Results and Discussion. How reflective of posterior, distal limb bud this cell line is more broadly could be explored in a future publication.

The cell line that we developed reflects distal limb mesenchyme but, as for any cell line, it is not equivalent. This cell line expresses a number of the genes found in the distal limb and the aspect that is important for our studies is that it retains a poised ZRS. Interestingly these lines retain limb identity since we can direct these cells to differentiate toward chondrocytes or tenocytes depending on culture conditions. So as the referee suggests we will attempt to publish these cells in a future publication, but for the purpose of the paper we have placed less emphasis on our lines as a model system for distal limb mesenchyme. In rewriting and shortening the Discussion I removed the reference that the cell line was a good model system.

Also, the section title has been changed to – “A limb-derived cell line shows ZRS activation and *Shh* induction”.

In Figure 1, the fold enrichment of H3K27ac at the ZRS is not so strong (ca. 2 fold) in the limb bud that contains ZPA, but much stronger enrichment is seen in control of the cultured cells (more than 10 fold in Figure 2). How is this difference explained? It would be thought that acetylation of H3K27 at the ZRS in ZPA is abolished under the culture condition due to detachment of AER signals. Is it not the case? Direct comparison of ChIP-qPCR for H3K27ac between in vivo and the culture condition may be one way to address this question. Alternatively, a time course of deacetylation at H3K27 in ZRS in cultured limb bud cells should be presented.

ChIP experiments are notoriously sensitive to a number of factors and quality of the antibody used appears most crucial. The differences detected between fold enrichments in limb buds (Figure 1) and cell culture (Figure 2) is more than likely due to the fact that we changed our supplier of the anti-H3K27ac antibody (from Abcam to Active Motif) during these studies. As the referee suggested we did a direct comparison of limb vs. cell culture in the same experiment and found very good agreement in the fold enrichment. These data are shown in the supplementary figures (Figure 2—figure supplement 3). In the text, we added the following comment…“The fold enrichment of H3K27ac in the limb buds [Figure 1] and cell lines [Figure 2] seemed dramatically different; therefore, enrichment of H3K27ac was directly compared between the limbs buds and the cells in the same experiment showing that the magnitude of enrichment is comparable [Figure 2—figure supplement 3]”.

Procedure of the 4C-seq is not clearly stated, and therefore the present result is not scientifically sound. The anchor (bait) point is not clearly shown. Digestion efficiency with restriction enzymes is not evaluated. Method of the 4C-DNA mapping is not well described. Mapping rate per total read counts is also lacking. Another concern is weak and very specific interaction between the ZRS and Shh. In a previous report from the same group (Anderson et al., 2014), it was stated that the Shh locus forms a topological domain where various genomic regions frequently interact with each other, consistent with Hi-C data from other groups. In Figure 3, ZRS seems to exclusively interact with Shh. It is an intriguing possibility that the TSA treatment facilitate a specific interaction in cultured cells. If it is the case, reproducibility of such weak interaction should be shown by experimental replicates of the 4C-analysis. Alternatively, 3C-qPCR could be fine to evaluate such specific interactions.

We have now added to this analysis, using 3C-qPCR, to confirm the significant increase in the association of the ZRS enhancer and *Shh* gene after activation of *Shh* expression in the cell line. We have added the statement … “The interaction between the *Shh* gene and the ZRS was confirmed by 3C-qPCR (Figure 3—figure supplement 1).”

The data has been added as a supplementary figure – Figure 3—figure supplement 1. The methodology for both the 3C-qPCR and 4C-seq is detailed in the Materials and methods and the digestion efficiencies and the ligation reactions are displayed in the supplementary figures (Figure 2—figure supplement 3, Figure 3—figure supplement 1). In addition for the 3C-qPCR technique the digestion efficiencies were assayed by PCR and the primers used are in TABLE S2. We now feel that using two different 3C technologies to analyze the enhancer promoter interactions helps to strongly support our conclusions that conformational changes occur in the cell line that brings the enhancer and promoter in contact and is acting in a similar fashion to the limb bud.

Figure 4 and legend, it is unclear if a specific region is analyzed (the ZRS? 5', 3'?). As presented it appears that H3K4me1 ChIP is severely altered at all loci by FGF inhibition with nintedanib. Presumably this is data from a single locus but it does lead to the question as to whether the tissue is severely compromised/dying. The relative expression of Grem1 goes up only modestly and relying on proximal tissue results is a biased comparison of tissue health since these cells are more mature.

We have now clarified in Figure 4) the primer pairs that were used in the experiment (3’LR) which is now stated in the figures. Also we now have given the reference of the mouse limb explant culture technique (Havis et al. [2014] Development 141, p3683) using FGFs and FGF inhibitors that we modified in our study which is now stated in the Materials and methods “Limb buds were dissected from E11.5 mouse embryos and cultured as described by Havis E. et al. (2014) with the following modifications”. We also tested for cell death in the culture treatment after 4 hrs using whole limb buds and found no extensive damage over the 4hr culture period (Figure 4—figure supplement 1). We state “Trypan blue staining was performed on limb dissections by following a reported protocol with slight modifications (Siddique Y.H 2012)(Figure 4—figure supplement 1). No signs of increased cell death was observed after 4h of NIN treatment”.

Etv4 null mutant mouse is viable, and exhibits no apparent defect in the limb morphology (Laing et al., 2000). This is inconsistent with the claim of this study that binding of ETV4 to the ZRS suppresses Shh. Is Etv4 indispensable for suppression of Shh in the fp14 cells, whereas in general ETV4 and other ETV proteins including ETV5 are functionally redundant. At a minimum the authors need to include a discussion about the possibility that other ETV proteins suppress Shh expression. It is preferable that the expression level of Etv5 in the cell culture conditions is analyzed.

In the 14Fp cell line the ETV5 mRNA levels are low and are some 8-10 fold lower than those for the ETV4 gene as shown in the supplementary figure (Figure 2—figure supplement 1); whereas, in distal limb mesenchyme there is less than a 2 fold difference. Thus ETV5 did not play a large role in our analysis. We have now added a sentence “Since in the 14Fp cell line ETV5 expression is low (approx. 8-10 fold lower than ETV4) (Figure 2—figure supplement 1), depletion of ETV5 activity was not necessary in this analysis.”

There is an opportunity to better summarize and highlight the contribution of this study in the conclusions; it currently reads more as a summary of results, both current and past, and could be shortened.

The conclusions have now been shortened and emphasis has been placed on the contribution of our studies.